# Bridging Offline and Online Experimentation: Constraint Active Search for Deployed Performance Optimization

**Junpei Komiyama**                                                    *junpei@komiyama.info*
*New York University*

**Gustavo Malkomes**                                          *gustavo.malkomes@intel.com*
**Bolong Cheng**                                                    *harvey.cheng@intel.com*
**Michael McCourt**                                            *michael.mccourt@intel.com*
*Intel SigOpt*

**Reviewed on OpenReview:** *https://openreview.net/forum?id=XX8CEN815d*

## Abstract

A common challenge in machine learning model development is that models perform differently between the offline development phase and the eventual deployment phase. Fundamentally, the goal of such a model is to maximize performance during deployment, but such performance cannot be measured offline. As such, we propose to augment the standard offline sample efficient hyperparameter optimization to instead search offline for a diverse set of models which can have potentially superior online performance. To this end, we utilize Constraint Active Search to identify such a diverse set of models, and we study their online performance using a variant of Best Arm Identification to select the best model for deployment. The key contribution of this article is the theoretical analysis of the two-phase development strategy, both in analyzing the probability of improvement over the baseline as well as the number of viable treatments for online testing. We demonstrate the viability of this strategy on synthetic examples, as well as a recommendation system benchmark.

## 1 Introduction

Developing a machine learning model for production requires a series of modeling decisions, e.g., the choice of data, loss function, model structure, and regularization. These choices may be broadly referred to as the "hyperparameters" of an ML model, and any choice of them will yield a viable model. An appropriate or optimal selection of them is often guided by loss or accuracy achieved on a validation set which is disjoint from the training set. The purpose of this validation objective is to avoid overfitting to the training loss by estimating the model performance on unseen data (Bishop, 2007).

There remains, however, a gap even between performance in an offline validation setting and online performance. Such a gap is well-known in industrial ML settings, where deployed models often fail to live up to the expectations set during an offline development process. Most notably, for our purposes, Krauth et al. (2020) states, after extensive empirical testing on recommender systems, that

> ... offline metrics are correlated with online performance over a range of environments. However, improvements in offline metrics lead to diminishing returns in online performance.

Radlinski et al. (2008) showed an example where the improvement of the click-through rate deteriorates the quality of a search engine. Kato et al. (2020) calls out the need to adapt offline models to account for the current network state as a key component in advancing wireless technology. The offline/online gap is even more prominent in circumstances where deployed ML systems can produce unphysical outcomes which must be avoided (Brenowitz et al., 2020).

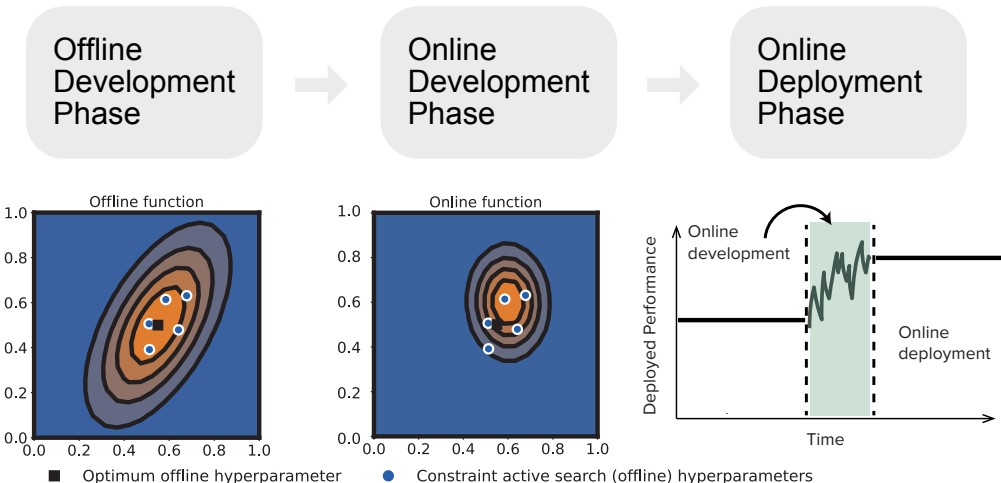

Figure 1: A graphical depiction of how our two-phase strategy precedes a deployment of a new ML model. In the offline development phase we explore an offline metric to find viable models for subsequent testing online, the best performing of which is deployed long-term.

In acknowledgment of these gaps, naive use of sample-efficient offline optimization tools, such as Bayesian optimization (Jones et al., 1998; Frazier, 2018), can lead to underperformance in an online setting. To address this issue, strategies have been developed for incorporating supplemental information (including online information) into the model development process (Swersky et al., 2013). Special tooling has been developed to support monitoring of deployed ML systems to utilize metrics recorded deployment (both model performance and contextual / environmental drift) for subsequent redesign of the ML model (Banerjee et al., 2020).

State-space models such as the Kalman filter have been developed to allow for data assimilation: models are developed offline but later altered or updated using online (real-time) information to improve their accuracy (Wan & Van Der Merwe, 2000; Kwiatkowski & Mandel, 2015). Chen et al. (2017) used offline data to modify the design space (decision variables) and allow their online genetic algorithm to more efficiently maintain the high performance of a reservoir system. These strategies, however, are in pursuit of a continually/iteratively run online tracking/optimization process.

In this article, we consider the setting where we have only a fixed window to conduct online development. Of course, the topic of online performance optimization has been addressed by many researchers. Multi-armed bandits (Robbins, 1952; Lai & Robbins, 1985; Komiyama et al., 2015) provide a strategy for constantly testing multiple treatments to minimize cumulative regret during the testing phase (which could extend indefinitely). Best arm identification (BAI) (Audibert et al., 2010) is used to measure the performance of multiple strategies online and choose one for eventual long-term deployment. We consider only the BAI circumstances (ignoring cumulative regret during online development/testing), as they most closely match our deployed circumstances.

Our proposed strategy for improving deployed performance works in two phases: an offline and an online development phase. First, during the offline development phase, we replace the standard hyperparameter optimization that eventually outputs a single model with Constraint Active Search (Malkomes et al., 2021) to create a diverse set of models with high offline performance. Second, we conduct a subsequent online development phase using a BAI-inspired algorithm (Algorithm 2 in Section 2.2) to identify the one model which performs best in an online setting. This model is then deployed, by itself, for a long period of time, during which the performance will ultimately be judged. This process is depicted in Figure 1.

---

**Algorithm 1** The two-phase development process

---

Run the offline development algorithm (CAS) to select $K$ models.
Run the online development algorithm (Algorithm 2) to select one of the $K$ models. This model will be used during the online deployment phase.

---

## 2 Proposed two-phase development strategy for online performance optimization

This strategy is closest to that of Letham & Bakshy (2019), who built multi-task Gaussian processes to incorporate offline (simulator) data into a sample-efficient online policy search. The authors considered a sophisticated scenario where offline and online observations can be collected in multiple successions. In general, we do not enjoy such luxury of successive alternating between offline/online experimentation using simulators. In this work, we consider a different setting that captures a more standard industry scenario where developers build offline models and, then only test these models in a short online development phase. We assume that the cost of online experimentation is expensive, where system administrators have to redirect the online traffic to perform experimentation. Notice that even if the operation aspect of managing online traffic is automated, there is an important business decision of not serving users with your current production model, which invariably comes with the risk of sacrificing performance.

In effect, this is a *one-shot* online optimization. See Paleyes et al. (2020) for some analysis of these circumstances. To define our setting and notation, during the two-phase process (Algorithm 1) we search for configurations $\boldsymbol{x}$ in a search space $\mathcal{X}$; in principle, this space could include categorical parameters, but for ease of exposition, we assume that $\mathcal{X} \subset \mathbb{R}^d$. In particular, we let $y(\boldsymbol{x}) : \mathcal{X} \to \mathbb{R}$ denote the expected deployed performance of the model $\boldsymbol{x}$ in the testing phase and $y_0$ be the expected baseline deployed performance; these expectations are taken over the deployed circumstances. We use $\tilde{y} : \mathcal{X} \to \mathbb{R}$ to denote the offline performance metric.

In our analysis, we primarily assume that the online metric $y(\boldsymbol{x})$ is the probability of a random user receiving a successful recommendation from model $\boldsymbol{x}$. No assumption is made regarding the structure of $\tilde{y}$ except that it is somewhat correlated to $y$ (so studying $\tilde{y}$ offline gives us some soft insights about $y$ online).

Our motivation in designing our strategy is to maximize the probability of the next deployed model $\boldsymbol{x}^{(T)}$ improving over the baseline,

$$\mathbb{P}\left[y\left(\boldsymbol{x}^{(T)}\right) > y_0\right].$$

Specifically, we consider a two stage-process such that: first, we use $M$ offline rounds to select $\boldsymbol{x}_1, \boldsymbol{x}_2, \ldots, \boldsymbol{x}_K$ candidate models based on the offline observations $\tilde{y}(\boldsymbol{x})$. In these offline settings, users may use any (or multiple) sources of data to compute $\tilde{y}(\boldsymbol{x})$. Recall that we make no specific structural assumptions about how $y$ and $\tilde{y}$ are correlated, but it is easy to imagine that $\tilde{y}(\boldsymbol{x})$ is a corrupted observation of $y(\boldsymbol{x})$. Then, we promote these $K$ candidate models (or arms) to an online development phase that will last $T$ rounds. At each online development iteration, we must choose which arm will receive a sample of $y\left(\boldsymbol{x}_{I(t)}\right)$, representing an online feedback of a single user, and where $I(t)$ denotes the index of the selected model at the development online iteration $t < T$. At termination, we expect that the last selected model $\boldsymbol{x}^{(T)}$ will improve the baseline performance, $\mathbb{P}\left[y\left(\boldsymbol{x}^{(T)}\right) > y_0\right]$, and will be used for deployment.

Maximization is used here for simplicity, and this could be easily rephrased as minimization. Here, the probability is defined over the space of $y$ and $\tilde{y}$ functions, as well as the random outcomes from the offline and online development phases. Next, we give more details about each proposed phase.

### 2.1 Offline development phase

During the offline development phase, we allot a budget of $M$ total offline model trainings in which to find $K$ candidates $\boldsymbol{x}_1, \boldsymbol{x}_2, \ldots, \boldsymbol{x}_K$. These will subsequently be tested online to find one candidate $\boldsymbol{x}^{(T)}$ for long-term deployment. Unfortunately, during the offline development phase, we lack access to $y$ and can only evaluate $\tilde{y}$.

We assume that $\tilde{y}$ is expensive to evaluate, and thus searching the space $\boldsymbol{X}$ for high-performing outcomes must be done in a sample efficient fashion. For such a problem, random search (Bergstra & Bengio, 2012)

and grid search provide simple optimization strategies, while Bayesian optimization (BO) (Mockus, 1994; Frazier, 2018) provides significantly stronger performance (Turner et al., 2021). These optimization strategies are powerful for optimizing $\tilde{y}$, but they ignore the fact that we actually hope to optimize $y$. As a result, any $K$ candidates selected from that optimization process were found without a specific goal of producing diverse outcomes.

The first key contribution of this work consists of proposing the use of Constraint Active Search (CAS) during this offline development phase (Malkomes et al., 2021). Crucially, this problem formulation was developed to find a diverse set of satisfactory outcomes rather than simply optimizing; that makes it ideal for using $\tilde{y}$ to define satisfactory outcomes on which $y$ will later be measured. We denote the performance threshold as $\tau$, thus defining the satisfactory region as

$$\mathcal{S} = \{\boldsymbol{x} \,|\, \tilde{y}(\boldsymbol{x}) \geq \tau\}.$$

We use the Expected Coverage Increase algorithm proposed in the original paper for solving CAS and give a more detailed description in Appendix A. After CAS has been run for $M$ iterations on $\tilde{y}$ with threshold $\tau$, we have found some number, $J$, of satisfactory models: $\boldsymbol{x}'_1, \ldots, \boldsymbol{x}'_J \in \mathcal{S}$. If, unfortunately, $J < K$, then either the $\tau$ value may be reconsidered to include more models, $M$ could be increased to try to find more models, or fewer than $K$ models can be considered for the online development phase. The more likely case is that $J \geq K$, meaning that we must *subselect* $K$ candidate models $\boldsymbol{x}_1, \ldots, \boldsymbol{x}_K \in \mathcal{S}$ for use in the online development phase. We use a strategy based on $K$-determinantal point processes ($K$-DPP) which uses the learned covariance from the CAS to disperse points in $\mathcal{S}$; more information is provided in Appendix B.

## 2.2 Online development phase

In the online development phase, we run a version of best arm identification (BAI, Audibert et al. 2010) with $T$ total samples provided generated during the offline development phase. Algorithm 2 describes our strategy. For each round $t = 1, 2, \ldots, T$, we choose an arm $\boldsymbol{x}_{I(t)}$ and receive a corresponding reward[1]

$$X(t) \sim \text{Bernoulli}\left(y\left(\boldsymbol{x}_{I(t)}\right)\right), \tag{1}$$

The notation $[\ell]$ is used to mean $[\ell] \equiv \{1, 2, \ldots, \ell\}$. We define the empirical mean as

$$\hat{y}_i(t) = \frac{1}{N_i(t)} \sum_{s < t : I(s) = i} X(s),$$

where $N_i(t)$ is the number of draws on arm $i$ at the beginning of round $t$. Although our problem is similar to the BAI problem, a notable difference from the best arm identification problem is that the control arm (arm 0) with performance $y_0$ is considered deterministic; our goal is to find an arm with expected performance larger than $y_0$. Our algorithm is inspired by the upper confidence bound algorithm (Lai & Robbins, 1985; Auer et al., 2002) as well as the recent good arm identification algorithm (Kano et al., 2019) and the thresholding bandit algorithm (Locatelli et al., 2016). While good arm identification and thresholding bandit problems are aimed at finding more than one arm that is above/below the threshold, we are interested in finding at least one arm above the threshold. Although the general algorithm proposed by Katz-Samuels & Jamieson (2020) can be used for this purpose, it involves phases ($l = 1, 2, \ldots$) where the early phases are not very sample-efficient at the cost of generality. Our algorithm does not involve such a phase and is more effective and simple for our goal.

Algorithm 2 produces $L_i(t)$ and $U_i(t)$, which are the lower and upper confidence bounds, respectively of $y(\boldsymbol{x}_i)$ by using the samples until round $t$. Here, we define $U_i(t) = 1$ and $L_i(t) = 0$ when $N_i(t) = 0$. Conceptually, our algorithm is simple. At each round, we draw an arm based on the upper confidence bound. Using the confidence region $[L_i(t), U_i(t)]$, we discard the arms that are unlikely to be the best. At the end of round $T$, we choose the most sampled arm $\boldsymbol{x}^{(T)} \in [K]$.

---

[1]Namely, we assume that each round gives a noisy estimator of $y$.

---

**Algorithm 2** Thresholded successive elimination

---

**Require:** Arms $\boldsymbol{x}_1, \ldots, \boldsymbol{x}_K$, # of Rounds $T$, threshold $y_0$, confidence level $\eta \in (0, 1)$.

    Initialize the upper confidence bound $U_i(0) = 1$ for each $i \in [K]$.

    Initialize the active set $\Psi(1) = \{\boldsymbol{x}_1, \ldots, \boldsymbol{x}_K\}$.

    **for** Each round $t = 1, 2, \ldots, T$ **do**

        Draw arm $\boldsymbol{x}_{I(t)} : I(t) = \arg\max_{\boldsymbol{x}_i \in \Psi(t)} U_i(t-1)$.

        Update the lower bound $L_i(t) = \hat{y}_i(t) - \sqrt{\frac{\log(N_i(t)^2/\eta)}{2N_i(t)}}$, for $\boldsymbol{x}_i \in \Psi(t)$.

        Update the upper bound $U_i(t) = \hat{y}_i(t) + \sqrt{\frac{\log(N_i(t)^2/\eta)}{2N_i(t)}}$, for $\boldsymbol{x}_i \in \Psi(t)$.

        Update the active set $\Psi(t+1) = \left\{i \in [K] | U_i(t) \geq y_0, U_i(t) \geq \max_{j \in [K]} L_j(t), \right\}$.

    **end for**

    Output $\boldsymbol{x}^{(T)} = \arg\max_i N_i(T)$.

---

## 3 Theoretical analysis of our strategy

In this section, we provide some analysis of the components of our algorithm under reasonable assumptions regarding the development/deployment conditions.

### 3.1 Theoretical analysis of subselection of models

We assume the satisfactory region $\mathcal{S}$ is a compact subset of $\mathcal{X}$ that satisfies the following assumptions.

**Assumption 1.** (Lipschitz continuity) There exists a constant $L > 0$ such that

$$\sup_{\boldsymbol{x}, \boldsymbol{x}' \in \mathcal{X}} \frac{|y(\boldsymbol{x}) - y(\boldsymbol{x}')|}{|\boldsymbol{x} - \boldsymbol{x}'|} \leq L.$$

**Assumption 2.** (Existence of improvement) There exists $\boldsymbol{x}^* \in \mathcal{S}$ such that

$$y(\boldsymbol{x}^*) > y_0.$$

Let $y^* = y(\boldsymbol{x}^*)$ and $D = y^* - y_0$.

Note that, if we assume $y$ to be a sample path of the Gaussian process, then with a high probability, the continuity assumption (Assumption 1) is satisfied (c.f., Theorem 2 in Srinivas et al. 2012).

Assumption 2 is very mild; it only requires at least one point of improvement in the satisfactory region. Of course, if no such point exists, then either the modeling process must be improved (a common fear from all modelers), a different $\tilde{y}$ must be defined to more strongly correspond to $y$, or $\tau$ must be adjusted to include such a point.

**Remark 1.** (Volume of the region of improvement) Let

$$\mathcal{S}_{\text{imp}} := \{\boldsymbol{x} \in \mathcal{S} : y(\boldsymbol{x}) > y_0\},$$

and $\text{Vol}(\mathcal{S}_{\text{imp}})$ be the volume of the region of improvement.

**Assumption 3.** (Positive constant-ratio covering number) For any radius $r > 0$ there exists a covering $\boldsymbol{x}_1, \boldsymbol{x}_2, \ldots, \boldsymbol{x}_{\xi(r)}$ such that (1) any point $\boldsymbol{x} \in \mathcal{S}$ is covered:

$$\forall_{x \in \mathcal{S}} \exists_{i \in [\boldsymbol{x}_{\xi(r)}]} |\boldsymbol{x} - \boldsymbol{x}_i| \leq r,$$

(2) and the minimum ratio of volume is positive:

$$C_{\text{Cov}}(r) = \min_{i, j \in [\xi(r)]} \frac{\text{Vol}(\{\boldsymbol{x}' \in \mathcal{S} : |\boldsymbol{x}' - \boldsymbol{x}_i| \leq r\})}{\text{Vol}(\{\boldsymbol{x}' \in \mathcal{S} : |\boldsymbol{x}' - \boldsymbol{x}_j| \leq r\})} > 0.$$

Assumptions 1–3 imply that the volume of improvement is positive. In the proof of Theorem 1, we use these assumptions to lower-bound $\mathrm{Vol}(\mathcal{S}_{\mathrm{imp}})$.

**Remark 2.** (Counterexample of constant-ratio covering) By the assumption of the compactness of $\mathcal{S}$, the covering number always exists. The assumption on the constant-ratio covering with a positive $C_{\mathrm{Cov}}$ is very mild,[2] but we can bring some counterexamples. An obvious counterexample is that the volume $\mathcal{S}$ involves a single isolated point,[3] which is a limit of a very sharp shape illustrated in Appendix C where $C_{\mathrm{Cov}}$ can be arbitrarily small.

Given these assumptions, the quality of the best solution among the random sample from the satisfactory region is bounded as defined in Theorem 1.

**Theorem 1.** (Quality of random samples from the satisfactory region) Let $\boldsymbol{x}_1, \boldsymbol{x}_2, \ldots, \boldsymbol{x}_K$ be i.i.d. samples from $\mathcal{S}$. Then, there exists a model-dependent constant $C_{\mathrm{model}} > 0$ such that

$$\mathbb{P}\left[\max_i y(\boldsymbol{x}_i) \geq y_0\right] \geq 1 - \delta$$

for $K \geq C_{\mathrm{model}} \log(1/\delta)$.

See Appendix D for the proof of this theorem.

**Remark 3.** (I.i.d. sampling required in Theorem 1) This proof requires that $\boldsymbol{x}_1, \boldsymbol{x}_2, \ldots, \boldsymbol{x}_K$ be drawn i.i.d. from $\mathcal{S}$. However, in practice, it is challenging to sample i.i.d. points from $\mathcal{S}$. We adopted the $K$-DPP strategy (Appendix B) that disperses points in a max-cover fashion. We consider that this is a reasonable approximation to i.i.d. sampling.

**Remark 4.** (Minimum bound on $K$) The value $K$ appears in Theorem 1 that guarantees the improvement with probability $1 - \delta$ is dependent on the model. Unfortunately, knowing this bound is functionally impossible because the full knowledge of $y$ on $\mathcal{S}$ (i.e., the performance of the model in the online development phase) is unknown in our setting. In Section 3.3, we further consider the choice of $K$ and identify that the online development phase places only a mild upper bound of $K < \sqrt{T}$. An implication of this is somewhat counterintuitive. It suggests that bringing many models to the online development phase to choose one of them rather than being selective in the offline development phase.

### 3.2 Theoretical analysis of online development phase

The following theorem analyzes the probability of improving over the deployed baseline performance $y_0$ when running Algorithm 2.

**Theorem 2.** (Bound on probability of improving over the deployed baseline) Assume that there exists at least one point $\boldsymbol{x}_i$ such that $y(\boldsymbol{x}_i) > y_0$. Let $\Delta = (\max_i y(x_i) - y_0)/2$. Then, for $T$ such that[4]

$$2 \times \sqrt{\frac{\log((T/K)^2/\eta)}{2(T/K)}} \leq \Delta, \tag{2}$$

we have

$$y(\boldsymbol{x}^{(T)}) > y_0$$

with probability at least $1 - (K\pi^2\eta)/6$.

See Appendix E for a proof of Theorem 2. Here, the factor $\pi^2/6$ in the probability is derived from the fact that $\sum_n 1/n^2 = \pi^2/6$.

**Remark 5.** (Randomness in Theorem 2) The probability distribution employed in Theorem 2 is taken with respect to the randomness of the rewards. Since Algorithm 2 is deterministic, $\boldsymbol{x}^{(T)}$ is deterministic when the reward sequence is fixed.

---

[2]For example, these assumptions are satisfied in the case $\mathcal{S}$ is a hypersphere or a hyperrectangle.
[3]In this case, $C_{\mathrm{Cov}}(r) = 0$ for some $r > 0$.
[4]$T = \tilde{O}(K/\Delta^2)$ suffices.

**Remark 6.** (Exponential convergence) Theorem 2 states that the minimum value of $K\pi^2\eta/6$ such that Eq. (2) holds is

$$\frac{\pi^2 T^2}{6K} \exp\left(-\frac{2T\Delta^2}{K}\right),$$

which decays exponentially to $T$ given other parameters. Additionally, if $\Delta$ can, somehow, be estimated *a priori*, we can use Eq. (2) as guidance to set a lower bound on $T$.

**Remark 7.** (Required exploration before pruning $\Psi$) The value $\eta$ provides a direct mechanism for increasing/decreasing $U_i$. Setting $0 < \eta \ll 1$ will add a large amount to $U_i$, which will have the effect of allowing all arms to remain in $\Psi$ even after some poor early performance. As $N_i$ grows, the performance $\hat{y}_i$ will eventually dominate the quantity. But our Bernoulli reward structure (and the resulting bounds on $\hat{y}_i$ that it implies) provides an opportunity to force exploration without defining an explicit initialization phase.

**Remark 8.** (Uniform confidence bounds) We note that the proof of Theorem 2 is based on event $\mathcal{B}$, which implies that all the confidence region is valid consistently over all arms and rounds. While such a bound is conservative, most of the theoretical analyses on MAB/BAI rely on such a bound (e.g., Auer et al. (2002); Gabillon et al. (2012)). The exponential rate of concentration (Remark 6) justifies the use of such a bound. We should also note that Algorithm 2 is free from any forced exploration/initialization phase. Early in the BAI process, the confidence region is quite massive, meaning that the online development process likely needs no initialization to perform effectively.

**Remark 9.** (Reward assumption) In Eq. (1), we assume the reward is Bernoulli and thus $\hat{y}_{i,n}$ unbiased estimator of $y(\boldsymbol{x}_i)$. Namely, $\mathbb{E}[\exp(t(X - y))] \leq \exp(R^2 t^2/2)$ for some parameter $R > 0$. It is easy to create a new algorithm by introducing the confidence bound of magnitude $R\sqrt{\frac{2\log(N_i(t)^2/\eta)}{N_i(t)}}$. This sub-Gaussian assumption is general enough to cover many discrete and continuous feedback, such as binary (e.g., click and non-click), ordered multinomial (e.g., 5-star rating), and Gaussian (e.g., numeric scores). For a structural model where single feedback $X(t)$ does not give an unbiased estimator of $y(\boldsymbol{x}_i)$, we can create a similar algorithm if we can build an anytime confidence bound that holds consistently over any number of samples (i.e., an equivalent to Event $\mathcal{B}$ in the proof of Theorem 2).

### 3.3 Implication of choice of K

We consider the relationship between $K$ and the probability of improving over the baseline. A large $K$ increases the probability of finding at least one improvement. On the other hand, a small $K$ increases the probability of actually returning $\boldsymbol{x}^{(T)} = \arg\max_{\boldsymbol{x}\in\{\boldsymbol{x}_1,\dots,\boldsymbol{x}_K\}} y(\boldsymbol{x})$. Namely, results in Section 3.1 state that

$$\mathbb{P}\left[\max_i y(\boldsymbol{x}_i) < y_0\right] = \left(1 - \frac{\text{Vol}(\mathcal{S}_{\text{imp}})}{\text{Vol}(\mathcal{S})}\right)^K = e^{-CK},$$

where $C = \log\left(\left(1 - \frac{\text{Vol}(\mathcal{S}_{\text{imp}})}{\text{Vol}(\mathcal{S})}\right)^{-1}\right)$. At the same time, Remark 6 states that

$$\mathbb{P}\left[y\left(\boldsymbol{x}^{(T)}\right) \geq y_0 \,\middle|\, \max_i y(\boldsymbol{x}_i) > y_0 + \Delta\right] = \frac{\pi^2 T^2}{6K} \exp\left(-\frac{2T\Delta^2}{K}\right).$$

Assuming that $\Delta > 0$ is a positive constant, the value of $K$ that minimizes the sum of these two terms is derived as

$$\arg\min_{K>0}\left(e^{-CK} + \frac{\pi^2 T^2}{6K} \exp\left(-\frac{2T\Delta^2}{K}\right)\right),$$

which demonstrates $K = \tilde{\Theta}\left(\sqrt{T}\right)$, where $\tilde{\Theta}$ is Landau notation that ignores a polylogarithmic factor.

There are several practical considerations related to the choice of $K$. First, the optimal choice of $K$ is model dependent; when the satisfactory region involves a very small volume of improvement $\text{Vol}(\mathcal{S}_{\text{imp}})$, it favors a large value of $K$. Meanwhile, a large value of $K$ indicates that we test many candidate models in the testing

phase, which not only decreases the chance of finding the best arm but also makes the online development process more stochastic.

Additionally, one result of the online development is a confidence interval on the expected performance of $\boldsymbol{x}^{(T)}$ during deployment – many industrial settings may enforce a maximum possible size of this confidence interval before deployment. Such a requirement could encourage smaller $K$ (assuming that $T$ is fixed). This is in addition to the CAS process which, because it often requires expensive training, is generally run on a limited budget and may only produce roughly $K$ satisfactory results (as opposed to the $\sqrt{T}$ which might be viable during online development).

## 4 Experimentation

We provide numerical experiments to demonstrate the consistency of our strategy on synthetic problems as well as the viability of our strategy in a recommender system benchmark. We have chosen $M$ and $T$ arbitrarily; in a standard industrial setting, $M$ might be chosen based on development deadlines and $T$ might be chosen to generate a desired confidence interval for the eventually deployed model. We fix $\eta = 0.01$ to require a minimum amount of exploration before eliminating arms. For CAS and BO, we fit a constant mean with generalized least squares and use a $C^4$ Matérn covariance kernel with process variance and length scales learned through maximum likelihood estimation (Fasshauer & McCourt, 2015). CAS uses expected coverage improvement (ECI) (Malkomes et al., 2021) as its acquisition function; BO uses expected improvement (Frazier, 2018).

As a baseline, we compare our strategy involving testing $K$ arms during the online development phase against a more standard choice of only one arm as identified through BO. We also consider the choice of $K$ arms, randomly sampled from $\mathcal{S}$ with UCB used during the online development phase (denoted as "RS+UCB"). The line "Best CAS" represents the arm identified during CAS which has the best online performance (independent of whether it was part of the $K$ subselection). The numerical results are presented in the form "mean (standard error)". Here, randomness is considered over the output of the full development process.

### 4.1 Synthetic experiments

To demonstrate the implications of our proposed strategy, we created 20 offline/online problems using variations of objective functions parameterized like Gaussians (representative functions are depicted in Figure 1). The online objectives vary slightly from the offline objectives.

The center of the offline Gaussian function is $(0.5, 0.5)$ with covariance $0.05 \times \mathbf{I}$. The center of the online Gaussian function shifted from $(0.5, 0.5)$ to a random direction by a predetermined size $\lambda \in \{0.01, 0.05, 0.1, 0.15, 0.2\}$; the complete definition of these objectives is shown in Appendix F. We set $\tau = 0.75$ (the function range is $[0, 1]$) and the CAS resolution parameter $r = 0.08$. We ran the online development phase for $T = 1000$ rounds.

Table 1: Demonstration of our two-phase strategy on synthetic problems with increasing levels of difference between $\tilde{y}$ and $y$; we have fixed $K = 4$, $y_0 = 0.7$, $\tau = 0.75$, $M = 25$, $T = 1000$. As expected, our strategy has a consistent improvement over the BO-based solution when $\tilde{y}$ and $y$ are rather different and a value $y(\boldsymbol{x}) > y$ exists for $\boldsymbol{x} \in \mathcal{S}$. The "$-$" results indicate that at least one replication failed to produce a successful arm during Algorithm 2, which should be expected given the Best CAS expected value $0.69 < y_0$.

| Selected Model | Magnitude shift between offline and online ($\lambda$) | | | | |
| --- | --- | --- | --- | --- | --- |
| | 0.01 | 0.05 | 0.10 | 0.15 | 0.20 |
| BO | *1.00 (0.00)* | *0.91 (0.01)* | 0.70 (0.02) | 0.46 (0.03) | 0.27 (0.04) |
| RS + UCB | 0.74 (0.01) | 0.76 (0.02) | 0.75 (0.02) | 0.68 (0.02) | 0.59 (0.02) |
| Our Strategy | 0.95 (0.01) | *0.91 (0.02)* | *0.84 (0.02)* | *0.77 (0.03)* | $-$ (−) |
| Best CAS | 0.95 (0.01) | 0.95 (0.01) | 0.91 (0.02) | 0.84 (0.03) | 0.69 (0.04) |

Table 2: Demonstration of our two-phase strategy on synthetic problems with varying choices of $K$; we have fixed $\lambda = 0.1$, $y_0 = 0.7$, $\tau = 0.75$, $M = 25$, $T = 1000$. The BO and Best CAS outcomes are independent of $K$, as represented by "..." in the table. As expected Remark 4, increasing $K$ has a strictly positive effect on online development performance (up to the stated threshold).

| | Number of models in online development ($K$) | | | |
|---|---|---|---|---|
| **Selected Model** | 2 | 3 | 4 | 5 |
| BO | 0.70 (0.02) | ... | ... | ... |
| RS + UCB | 0.70 (0.02) | 0.73 (0.02) | 0.75 (0.02) | 0.76 (0.02) |
| Our Strategy | *0.76 (0.03)* | *0.78 (0.03)* | *0.84 (0.02)* | *0.86 (0.02)* |
| Best CAS | 0.91 (0.02) | ... | ... | ... |

Table 1 shows the results for the set of experiments where we subselected $K = 4$ candidates. In the cases where offline and online objectives are very similar (smaller $\lambda$), the diversity provided by CAS slightly compromises the online performance. In the case where the offline and online objectives do not match perfectly, we expect our proposed strategies to outperform the best configuration achieved during the offline development phase. Table 2 shows the results with $K = 2, 3, 4, 5$. Testing more models in the online development phase increases the final online performance, even though we have fewer samples per arm. Additional results (varying $y_0$ and $\tau$) are presented in Appendix F; those results also suggest that our hypothesis that DPP subselection favors more diversity than ECI – we consider that diversity among the selected $K$ models improves the quality of the final model when the gap between the offline and online performances increases.

## 4.2 Recommender system experiment

Next, we consider the popular "Learning to rank" problem (Qin & Liu, 2013) as an example of recommender systems on which we test our strategy. Our recommendation system adopts XGBoost, which learns the map from the query-url joint features to the relevance score. Our goal is to choose hyperparameters of XGBoost to maximize online deployment performance according to the normalized discounted cumulative gain (NDCG), which rewards accurate rankings. In our offline development phase, we train XGBoost models using the training set and perform hyperparameter tuning on the average NDCG of the validation set; we denote this metric by $\tilde{y}$.

During the online development phase, we ran $T = 4000$ queries by uniformly selecting elements (with replacement) from the test set. For each query, we collect binary rewards corresponding to Bernoulli samples with a probability of success (the $y$ function) equal to the top-1 NDCG (NDCG@1) score of the candidate model prediction vs. test ranking. A perfect ranking will have a probability 1 of receiving a positive reward. Figure 2 provides some perspective on the relationship between $\tilde{y}$ and $y$.

Table 3: Demonstration of our two-phase strategy on the content ranking problem; we have fixed $K = 4$, $y_0 = 0.33$, $M = 100$, $T = 4000$. For MQ2008 we used $\tau = 0.57$, and for the MSLR-WEB10K datasets we used $\tau = 0.69$. We show *expected online deployment* results for both strategies across 100 repetitions. We highlight results that are significant with a confidence level of 0.001, according to a standard t-test.

| | Learning to rank dataset | | |
|---|---|---|---|
| **Selected Model** | MQ2008 | MSLR-WEB10K | MSLR-WEB10K + noise |
| BO (best offline) | 0.338 (0.001) | *0.438 (0.001)* | 0.433 (0.001) |
| Our Strategy | **0.350 (0.001)** | *0.438 (0.001)* | **0.436 (0.001)** |

We see in Figure 2 that there is enough correlation between $\tilde{y}$ and $y$ to define a satisfactory region. There is still some discrepancy between them within the satisfactory region of the MQ2008 dataset. However, the MSLR-WEB10K dataset shows a very high correlation between offline and online metrics, which means that

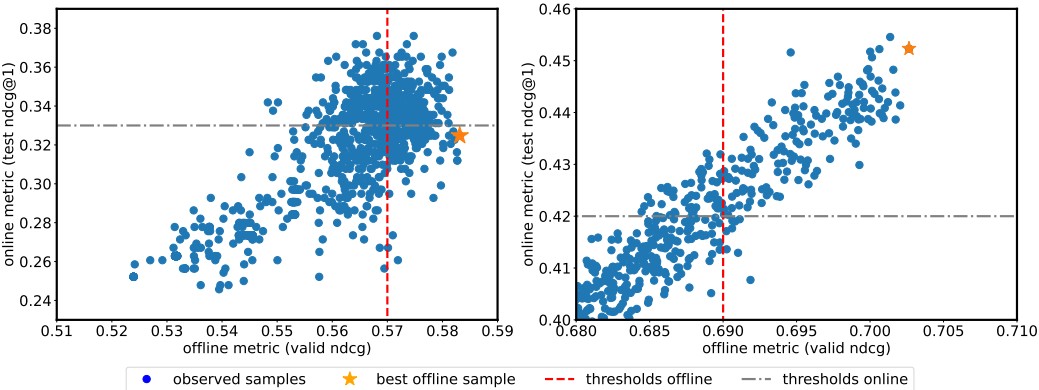

Figure 2: The $\tilde{y}$ and $y$ values in these learning to rank problems are clearly correlated in low performance models, but less strongly correlated in high performance models. The vertical dashed red line represents $\tau$ and the horizontal dashed black line represents $y_0$. *left:* MQ2008 dataset. *right:* MSLR-WEB10K dataset.

optimizing the offline metric by BO suffices for obtaining good online performance. To demonstrate our proposed method can be advantageous when there is a gap between offline and online metrics, we have also conducted experiments in a modified version of MSLR-WEB10K, where we add zero-mean Gaussian noise with a standard deviation of 0.01 to the offline metric, forcing $\tilde{y}$ and $y$ to be different.

Table 3 shows the performance of the selected model in the online deployment phase, where we have computed the top-1 NDCG score averaged over the entire test set to simulate the final deployment performance. As expected, when $\tilde{y}$ and $y$ do not perfectly match (datasets MQ2008 and MSLR-WEB10K with noise), our proposed approach provides a benefit over selecting the best offline model (chosen by using BO). Specifically, during the offline phase, CAS finds diverse high-performing models that are not optimal according to the offline metric, and Algorithm 2 uses the online development phase to identify which models are likely to improve our desired performance in the online deployment. Interestingly, even when both metrics are correlated, as in dataset MSLR-WEB10K, our proposed approach is as good as BO.

## 5 Conclusion

We have proposed a two-phase development strategy, depicted in Figure 1 for improving online deployed performance: first utilizing CAS during offline model development to produce $K$ viable options, which are then sent to Algorithm 2 for online model development to find a single model for long-term deployment. The theoretical analysis provides confidence about the outcome of the online development phase, and the numerical studies show the impact of possible decisions made in the offline development phase. As expected, our strategy provides a noticeable improvement over the previously deployed baseline in circumstances where the offline and online metrics are not perfectly correlated.

More work must be done on this topic to understand the implications of some of the free parameters in our implementation, among them $\eta$, $y_0$, $\tau$. We also would like to extend the theory to more effectively account for the non-i.i.d. nature of our $K$ arm selection as it affects Theorem 1. Our proposal here assumes only a single metric in the online setting, but we would like to be able to consider online performance with multiple metrics (CAS would naturally be able to handle this already in the offline setting). Also, Theorem 2 studies the probability of exceeding $y_0$ for a Bernoulli reward, but future work could extend this as described in Remark 9 as well as consider the magnitude of improvement (rather than the existence of improvement). We could also revisit the assumption of a fixed $y_0$ value and allow $y_0$ to be learned during the online development process using some fraction of $T$.

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

## A   Constraint Active Search

During the offline phase of our approach, we solve the Constraint Active Search (CAS) problem proposed by Malkomes et al. (2021). This formulation was developed with the goal of finding a diverse set of satisfactory points rather than optimizing a single objective function. Let $\tau$ be our performance threshold; then our goal is to search for the following satisfactory region

$$\mathcal{S} = \{\boldsymbol{x} \mid \tilde{y}(\boldsymbol{x}) \geq \tau\}.$$

We use the Expected Coverage Improvement (ECI) introduced in the original CAS publication Malkomes et al. (2021), which we summarize below for completeness.

Let's first assume we have a probabilistic model $\mathcal{M}$ that captures our prior belief about observations of $\tilde{y}(\boldsymbol{x})$. Any observational model should work, as long as we can perform posterior updates given the data $\mathcal{D}$, $\mathbb{P}(\tilde{y} \mid \mathcal{D})$, as well as the predictive distribution at any new location $\boldsymbol{x} \in \mathcal{X}$. For example, in our experiments we place a Gaussian Process over $\tilde{y}(.)$ with a standard additive Gaussian noise assumption. Let $Z$ be an indicator variable that tells us if a point $\boldsymbol{x}$ will satisfy the thresholds, $Z(\boldsymbol{x}) = \mathbf{1}\big[\tilde{y}(\boldsymbol{x}) \geq \tau\big]$. Our probabilistic model $\mathcal{M}$ should let us compute the probability of any point $\boldsymbol{x}$ belonging to the satisfactory region $\mathcal{S}$ after observing a dataset $\mathcal{D}$, which we denote by $\mathbb{P}(Z(\boldsymbol{x}) = 1 \mid \mathcal{D}, \mathcal{M})$.

A key parameter of ECI is the coverage radius $r$, which essentially captures the notation of diversity: points with a distance larger than $r$ are considered to be different, whereas points within $r$ are similar points. Intuitively, the algorithm seeks to "cover" the volume of $\mathcal{S}$ using a small number of "balls" of radius $r$.

**Definition 1** (Coverage neighborhood, Malkomes et al. (2021))**.** The *coverage neighborhood* of any $\boldsymbol{x}$ is defined as

$$\mathcal{N}_r(\boldsymbol{x}) = \{\boldsymbol{x}' : d(\boldsymbol{x}, \boldsymbol{x}') < r\},$$

for an *a priori* fixed $r \in \mathbb{R}^+$ and an appropriate distance function $d : \mathcal{X} \times \mathcal{X} \mapsto \mathbb{R}^+$. The coverage neighborhood of a set of points $\mathbf{X}$ is defined as

$$\mathcal{N}_r(\mathbf{X}) = \bigcup_{\boldsymbol{x} \in \mathbf{X}} \mathcal{N}_r(\boldsymbol{x}).$$

Note that, for Euclidean distance, $\mathcal{N}_r(\boldsymbol{x})$ is simply a ball of radius $r$.

Since the satisfactory region is unknown, we use a probabilistic model to compute the expected volume increase given by our prospective point $\boldsymbol{x}$. Letting $\mathcal{N}(\boldsymbol{x})$ be a finite approximation of the search space, ECI is summarized by the following acquisition function:

$$\begin{aligned}
\alpha(\boldsymbol{x} \mid \mathcal{D}) &= \mathbb{E}_Z\Bigg[\sum_{\boldsymbol{x}' \in \mathcal{N}(\boldsymbol{x}) \setminus \mathcal{N}_r(\mathbf{X})} Z(\boldsymbol{x}')\Bigg] \\
&= \sum_{\boldsymbol{x}' \in \mathcal{N}(\boldsymbol{x}) \setminus \mathcal{N}_r(\mathbf{X})} \mathbb{E}_Z\big[Z(\boldsymbol{x}')\big] \\
&= \sum_{\boldsymbol{x}' \in \mathcal{N}(\boldsymbol{x}) \setminus \mathcal{N}_r(\mathbf{X})} \mathbb{P}(Z(\boldsymbol{x}') = 1 \mid \mathcal{D}, \mathcal{M}).
\end{aligned}$$

which computes the expected new volume induced by a point $\boldsymbol{x}$ given the observed data $\mathcal{D}$, where $\mathbf{X}$ is just the input locations of the dataset $\mathcal{D}$. In practice, we approximate the expectation of $\mathbb{E}_Z$ by uniformly sampling points within $r$ of $\boldsymbol{x}$. In particular, we use BoTorch to efficiently compute ECI using Monte Carlo sampling (Balandat et al., 2020)[5].

The ECI algorithm is given by Algorithm 3. The algorithm outputs all the selected observations.

After running CAS for $M$ iterations on $\tilde{y}$ with threshold $\tau$, we have found some number, $J$, of satisfactory models: $\boldsymbol{x}'_1, \ldots, \boldsymbol{x}'_J \in \mathcal{S}$. If, unfortunately, $J < K$, then either the $\tau$ value may be reconsidered to include more models, $M$ could be increased to try to find more models, or fewer than $K$ models can be considered

---

[5]See this online tutorial: `https://botorch.org/v/0.6.2/tutorials/constraint_active_search`

---

**Algorithm 3** Constraint Active Search with Expected Coverage Increase

---

**Require:** Input space $\mathcal{X}$, threshold $\tau$, number of offline rounds $M$, diversity radius $r$, number of initial points $p$, objective function evaluator $\tilde{y}$, probabilistic model $\mathcal{M}$.
  Initialize the dataset $\mathcal{D}$ with $p$ random observations $\{\boldsymbol{x}_i, \tilde{y}(\boldsymbol{x}_i)\}_1^p$
  **for** Each iteration $m = p+1, p+2, \ldots, M$ **do**
     Update probabilistic model $\mathcal{M}$ using data $\mathcal{D}$
     $\boldsymbol{x}^* = \arg\max_{\boldsymbol{x} \in \mathcal{X}} \alpha(\boldsymbol{x}|\mathcal{D}) : \alpha(\boldsymbol{x}|\mathcal{D}) = \sum_{\boldsymbol{x}' \in \mathcal{N}(\boldsymbol{x}) \setminus \mathcal{N}_r(\mathbf{X})} \mathbb{P}(Z(\boldsymbol{x}') = 1 \mid \mathcal{D}, \mathcal{M})$
     $\mathcal{D} = \mathcal{D} \cup \{(\boldsymbol{x}^*, \tilde{y}(\boldsymbol{x}^*))\}$
  **end for**
  Output $\mathcal{D}$.

---

for the online development phase. The more likely case is that $J \geq K$, meaning that we must *subselect $K$* candidate models $\boldsymbol{x}_1, \ldots, \boldsymbol{x}_K \in \mathcal{S}$ for use in the online development phase. Next, we discuss two strategies for model subselection (Appendix B).

## B  K model subselection

During the offline development phase, $K$ models must be chosen from the $J > K$ models found during the CAS process. We could, simply, randomly choose $K$ models, but we propose two strategies that we hypothesize will be preferable.

**DPP subselection** During CAS, we fit a covariance kernel $\mathcal{K} : \mathcal{X} \times \mathcal{X} \to \mathbb{R}$ to the observed data as part of the search powered by the Gaussian process. Here, we reuse the learned kernel in our subselection strategy. In particular, we consider the $K$-DPP distribution (Kulesza & Taskar, 2011) defined over the Gaussian kernel $\mathcal{K}$ with the fixed set of $J$ points found during CAS. We return the mode of that distribution as our $K$-point subselection, i.e., the $K$ points for which $\det\left(\left(\mathcal{K}(\boldsymbol{x}_i', \boldsymbol{x}_j')\right)_{i,j \in [K]}\right)$ is maximized, where $\left(\mathcal{K}(\boldsymbol{x}_i', \boldsymbol{x}_j')\right)_{i,j \in [K]}$ denotes a $K \times K$ kernel matrix comprised of a size $K$ subset of points from the $J$ points found during CAS.

This subselection strategy is partly chosen to encourage these $K$ models to be widely spread throughout $\mathcal{S}$ – this provides us some capacity to approximate i.i.d. sampling from $\mathcal{S}$ which we analyze in Theorem 1. Rather than using a Euclidean metric (to spread points out in parameter space), we use the covariance kernel to also inform this process to incorporate any anisotropy of $\tilde{y}$ that we learn through CAS. Assuming that $\tilde{y}$ and $y$ belong to a similar reproducing kernel Hilbert space, a lower covariance implies a lower mutual dependence among $y(\boldsymbol{x}_1), y(\boldsymbol{x}_2), \ldots, y(\boldsymbol{x}_K)$, which maximizes the probability of finding at least one model that outperforms the baseline (i.e, $\mathbb{P}[\cup_{i \in K} \{y(\boldsymbol{x}_i) > y_0]\}$).

**ECI subselection** Another alternative is to greedily maximize the Expected Coverage Increase (ECI) (Malkomes et al., 2021). First, we add the satisfactory point with the highest $\tilde{y}$, then sequentially choose the other $K - 1$ candidates such that ECI is maximized. Similarly to the $K$-DPP strategy, we can use the Kernel distance.

## C  Coverings with and without constant ratio

We used an assumption about a constant covering ratio. Figure 3 illustrates the coverings with and without constant ratio. While it can, indeed, be the case that the satisfactory region takes such a shape, we believe that it is acceptable, in practice, to consider only the subset of $\mathcal{S}$ which has a constant covering ratio. This is a byproduct of the fact that, while such a $\mathcal{S}$ region may exist, we have only the ability to identify points in $\mathcal{S}$ through CAS. Therefore, our knowledge of that space (and subsequent ability to approximate i.i.d. sampling within that space) will logically also have this constant covering property.

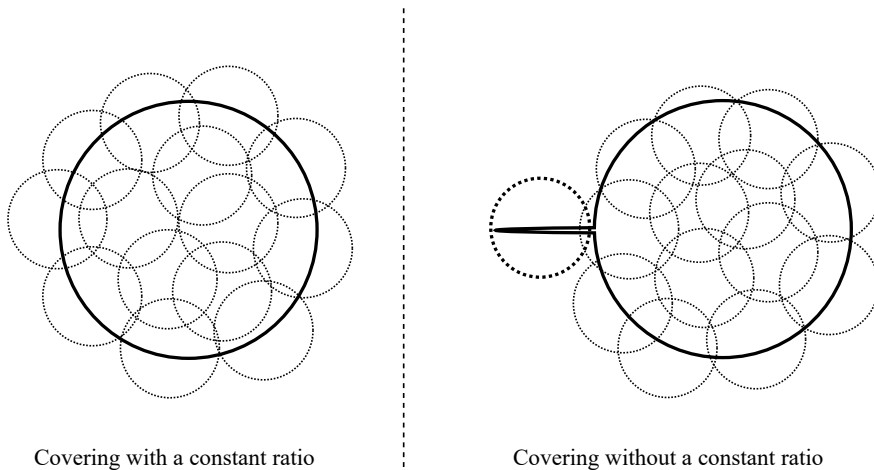

Covering with a constant ratio       Covering without a constant ratio

Figure 3: Illustrative example of covering with a constant ratio (left) and without a constant ratio (right). In the right figure the sharp spine has very little associated volume.

## D  Proof of Theorem 1

*Proof of Theorem 1.* First, we derive a lower bound on $\text{Vol}(\mathcal{S}_{\text{imp}})$. By Assumption 1 we have

$$\sup_{\boldsymbol{x}, \boldsymbol{x}' \in \mathcal{X}} \frac{|y(\boldsymbol{x}) - y(\boldsymbol{x}')|}{|\boldsymbol{x} - \boldsymbol{x}'|} \leq L. \tag{3}$$

Let $\mathcal{N}_{x^*}$ be a ball of radius $D/L$ centered at $\boldsymbol{x}^*$. Then, for any point $\boldsymbol{x} \in \mathcal{N}_{x^*}$, we have

$$
\begin{aligned}
y(\boldsymbol{x}) &\geq y(\boldsymbol{x}^*) - L \times \frac{D}{L} \quad \text{(by (3))} \\
&= y(\boldsymbol{x}^*) - D \\
&\geq y_0 \quad \text{(by Assumption 2)},
\end{aligned}
$$

which implies $\mathcal{N}_{\boldsymbol{x}^*} \subset \mathcal{S}_{\text{imp}}$. Let $\mathcal{N}_1, \mathcal{N}_2, \ldots, \mathcal{N}_{\xi(D/(2L))}$ be the covering of $\mathcal{S}$ with $r = D/(2L)$ in Assumption 3, with their centers $\boldsymbol{x}_1, \boldsymbol{x}_2, \ldots, \boldsymbol{x}_{\xi(D/(2L))}$, respectively.[6] Without loss of generality, assume that $\mathcal{N}_1$ includes $\boldsymbol{x}^*$. We have

$$
\begin{aligned}
\frac{\text{Vol}(\mathcal{S}_{\text{imp}})}{\text{Vol}(\mathcal{S})} &\geq \frac{\text{Vol}(\mathcal{N}_1)}{\text{Vol}(\mathcal{S})} \quad \text{(by } \mathcal{N}_1 \subset \mathcal{N}_{x^*} \subset \mathcal{S}_{\text{imp}}) \\
&\geq \frac{\text{Vol}(\mathcal{N}_1)}{\sum_{i=1}^{\xi(D/(2L))} \text{Vol}(\mathcal{N}_i)} \\
&\geq \frac{C_{\text{Cov}}(D/(2L))}{\xi(D/(2L))}, \quad \text{(by Assumption 3)}
\end{aligned} \tag{4}
$$

where $\mathcal{N}_1 \subset \mathcal{N}_{\boldsymbol{x}^*} \subset \mathcal{S}_{\text{imp}}$ holds because the distance between any two points in a ball of radius $D/(2L)$ is at most $D/L$. Eq. (4) provides a lower bound on $\text{Vol}(\mathcal{S}_{\text{imp}})$.

Since we draw i.i.d. samples from $\mathcal{S}$, we have

$$\mathbb{P}\left[\max_i y(\boldsymbol{x}_i) \geq y_0\right] = 1 - \left(1 - \frac{\text{Vol}(\mathcal{S}_{\text{imp}})}{\text{Vol}(\mathcal{S})}\right)^K. \tag{5}$$

Eq.(4) and (5) implies that, if we have

$$K \geq \frac{\log(\delta^{-1})}{\log\left(\left(1 - \frac{C_{\text{Cov}}(D/(2L))}{\xi(D/(2L))}\right)^{-1}\right)}, \tag{6}$$

---

[6]Here, the radius of each $\mathcal{N}_1, \mathcal{N}_2, \ldots, \mathcal{N}_{\xi(D/(2L))}$ is $D/2L$, whereas the radius of $\mathcal{N}_{\boldsymbol{x}^*}$ is $D/L$.

then we have

$$\left(1 - \frac{\text{Vol}(\mathcal{S}_{\text{imp}})}{\text{Vol}(\mathcal{S})}\right)^K \leq \delta.$$

In summary, with probability $1 - \delta$, we have the following.

$$\mathbb{P}\left[\max_i y(\boldsymbol{x}_i) \geq y_0\right] \geq 1 - \delta$$

for $K$ such that Eq (6) holds. □

## E   Proof of Theorem 2

*Proof of Theorem 2.* Let $\hat{y}_{i,n}$ be the value of $\hat{y}_i(t)$ when $N_i(t) = n$. Without a loss of generality, we assume arm 1 is the best arm (i.e., $1 = \arg\max_i y(\boldsymbol{x}_i)$). By assumption, we have $y(\boldsymbol{x}_1) > y_0$. Let $\Psi_{\text{sub}} = \{\boldsymbol{x}_i : y(\boldsymbol{x}_i) < y(\boldsymbol{x}_1) - \Delta\}$ be clearly suboptimal arms. Let

$$\mathcal{B} = \bigcap_{i \in [K]} \bigcap_{n \in [T]} \left\{ |\hat{y}_{i,n} - y(\boldsymbol{x}_i)| \leq \sqrt{\frac{\log(n^2/\eta)}{2n}} \right\}.$$

We have

$$\mathbb{P}[\mathcal{B}] \leq K \sum_n \frac{\eta}{n^2} \quad \text{(by Hoeffding inequality)}$$

$$\leq \frac{K\pi^2\eta}{6}.$$

Event $\mathcal{B}$ implies that

$$\bigcap_{i \in [K]} \bigcap_{t \in [T]} \{U_i(t) \geq y(\boldsymbol{x}_i) \geq L_i(t)\}. \tag{7}$$

Eq. (7) implies

$$U_1(t) \geq y(\boldsymbol{x}_1) \geq y(\boldsymbol{x}_i) \geq L_i(t)$$

for all $t$ and $i$, and thus arm 1 is never eliminated.

Let $i : \boldsymbol{x}_i \in \Psi_{\text{sub}}$ be arbitrary. Assume that $N_i(t) \geq T/K$. Then, for any $t' > t$, we have

$$U_1(t') \leq y(\boldsymbol{x}_i) + 2\sqrt{\frac{\log((N_i(t))^2/\eta)}{2N_i(t)}}$$

$$\text{(by Eq. (7) and } \log(n)/n \text{ is decreasing for } n \geq 3)$$

$$\leq y(\boldsymbol{x}_1)$$

$$\text{(by Eq. (2))}$$

$$\leq U_1(t'),$$

$$\text{(by Eq. (7))},$$

and thus arm $i$ is never drawn again. Therefore, at least one arm in $[K] \setminus \Psi_{\text{sub}}$ has been drawn more than $T/K$ times, and thus $\boldsymbol{x}^{(T)} \notin \Psi_{\text{sub}}$. □

## F   Ablation study on synthetic tests

In Section 4.1 we presented a synthetic experimental setting and considered the implications of different $\tilde{y}$ and $y$ functions on the eventual outcome. Here we present a more comprehensive analysis of the implications of different elements of our proposed two-phase strategy. Table 4 shows how the baseline performance which is currently deployed affects our strategy. Table 5 shows the impact of our $\tilde{y}$ offline performance threshold on the eventual online performance.

Table 4: We consider fixed experimental circumstances ($K = 4$, $\tau = 0.75$, $\lambda = 0.1$) and the impact of changing $y_0$, the currently deployed baseline. This has no impact on online optimization strategies like UCB, but it could impact Algorithm 2 because $y_0$ is incorporated into the online development process. Specifically, for $y_0 = 0.9$ all arms will be pruned because they do not improve the baseline. The BO and Best CAS outcomes are independent of $y_0$, as represented by "..." in the table.

| | Currently deployed baseline performance ($y_0$) | | | |
| Selected Model | 0.6 | 0.7 | 0.8 | 0.9 |
|---|---|---|---|---|
| BO | 0.72 (0.01) | ... | ... | ... |
| Our Strategy | 0.85 (0.01) | 0.85 (0.01) | 0.85 (0.01) | − (−) |
| Best CAS | 0.92 (0.01) | ... | ... | ... |

Table 5: We consider fixed experimental circumstances ($K = 4$, $y_0 = 0.5$, $\lambda = 0.1$) and the impact of changing $\tau$, the offline performance threshold defining $\mathcal{S}$. A small value for $\tau$ produces too much diversity and deteriorates its performance; very low values should recover random search. Large values of this quantity ($\geq 0.9$) should approximate CAS to BO alternative. The BO outcome is independent of $\tau$, as represented by "..." in the table.

| | Offline performance threshold ($\tau$) | | | |
| Selected Model | 0.3 | 0.5 | 0.75 | 0.9 |
|---|---|---|---|---|
| BO | 0.72 (0.01) | ... | ... | ... |
| Our Strategy | 0.68 (0.02) | 0.72 (0.02) | 0.85 (0.01) | 0.84 (0.01) |
| Best CAS | 0.91 (0.01) | 0.94 (0.00) | 0.92 (0.01) | 0.85 (0.01) |

# G   Experimental details for ranking problem

For the experiments in Section 4.2, we use the two datasets from the Microsoft "Learning to Rank" datasets: `MQ2008` and `MSLR-WEB10K`. We only use Fold 1 from each dataset and train XGBoost models on the training set for the ranking task. We only preprocessed the data by removing the query IDs from the features.

## G.1   Surrogate models for offline tuning

In order to reduce the computation time of repeated XGBoost training during the offline HPO process, we created two high-quality surrogate models for the HPO task. To create surrogate models, we first perform HPO (with random search) on XGBoost models where the hyperparameter search space presented in Table 6 and save all the corresponding XGBoost hyperparameter values (corresponding to $\boldsymbol{x}$) and the associated validation metric values (corresponding to $\tilde{y}$). For the `MSLR-WEB10K` dataset, we reduce the hyperparameter search space by removing `alpha` and `gamma`. The XGBoost models are trained to learn a relevant ranking and we adopt averaged NDCG as a validation metric.[7] We collected 1300 HPO results for each dataset.

Based on the HPO results, we fit an `ExtraTreesRegressor` (Scikit-learn) that models the function from the hyperparameter values to the validation metric so that can run offline HPO/CAS on the surrogate model very quickly. For example, training an XGBoost model on the MSLR-WEB10K takes approximately 5-10 minutes on the `c5.4xlarge` AWS instances[8] whereas drawing the corresponding value on the surrogate model requires less than one second.

---

[7]`https://xgboost.readthedocs.io/en/stable/parameter.html#learning-task-parameters`
[8]Hardware specifications of the C5 instances can be found at `https://aws.amazon.com/ec2/instance-types/c5/`

Table 6: XGBoost hyperparameter bounds for surrogate models

| Parameter Name | Type | Bounds | Log scaled |
|---:|:---:|:---:|:---:|
| alpha | double | $[0, 10]$ | None |
| eta | double | $[10^{-3}, 1]$ | True |
| gamma | double | $[0, 5]$ | None |
| lambda | double | $[0, 10]$ | True |
| max_delta_step | double | $[10^{-3}, 10]$ | None |
| max_depth | int | $[2, 16]$ | None |
| num_boost_round | int | $[10, 500]$ | None |

### G.2 Offline development phase with CAS

For the offline development phase, we created a wrapper around the BoTorch Expected Coverage Improvement implementation[9] to run constraint active search. We conduct CAS on the same hyperparameter search space as listed in Table 6. For both datasets, we choose a budget of 100, set the punchout radius $r = 0.15 \times \sqrt{\text{number of parameters}}$, and use $2 \times$ number of parameters initialization samples drawn from a Sobol sequence. We set the threshold to 0.57 for `MQ2008` dataset and 0.69 for `MSLR-WEB10K` respectively. These values represent approximately the 90th percentile of the offline evaluation metric (NDCG). For all experiments, we select four satisfactory models/arms using the K-DPP strategy outlined Section B; these models/arms will be used for the online phase of the experimentation.

For the BO baseline, we use the BoTorch Expected Improvement implementation. We use the same budget and initialization set up as the CAS experiments. We repeat the offline phase 100 times for CAS and BO.

### G.3 Online phase simulation

For the online development phase, we use Algorithm 2 with threshold $y_0 = 0.33$ for the `MQ2008` dataset and $y_0 = 0.42$ for the `MSLR-WEB10K` dataset. We fix $\eta = 0.5$ for both datasets. We did not use any surrogate model in this phase.

We ran $T = 4000$ queries by uniformly selecting elements (with replacement) from the test set. For each query, we collect Bernoulli samples with probability of success (true $y$) equal to the top-1 NDCG score of the candidate model prediction vs. test ranking. A perfect ranking will have a probability 1 of receiving a positive reward. Figure 2 provides some perspective on the relationship between $\tilde{y}$ and $y$.

We repeat the online development phase 10 times per offline CAS replication. For the first 100 iterations of the MAB simulation, we run a pure exploration policy before switching to Algorithm 2. Final results are computed using the average top-1 NDCG score across the entire dataset after selecting the models using MAB. We use the entire test set as a proxy for the online deployment performance of the system.

For the BO baselines, we take the best offline model/arm from each experiment (replication) and then compute the online deployment metric, top-1 NDCG, averaged over the entire test set. We then compute the mean and standard error of these metric values of online deployment.

### G.4 Note on compute resources

We conduct most of this experiment on AWS, using `C5.4xlarge` instances. We estimate the total CPU hours to be between 100 and 200 hours, and the most computationally intensive part is to build a surrogate model (c.f., Section G.1).

---

[9]BoTorch tutorial on CAS: `https://botorch.org/tutorials/constraint_active_search`

