# OpenReview forum: "Bridging Offline and Online Experimentation: Constraint Active Search for Deployed Performance Optimization"
_TMLR — Accepted by TMLR_

### Review · Reviewer_mPxg · 2022-08-14

**Summary Of Contributions:**

This paper proposes a two-stage model selection framework to bridge the performance gap between offline and online setting. This is motivated from a practical concern for model deployment in industry.

A model might perform well in offline experiment but its performance degrade during real-time, online testing because of mismatches between offline and online data distributions.

To address this, the proposed framework proceeds in two stages:

(1) offline -- running constraint-active-search (CAS) (Malkomes et al., 2021) which produces a list of K models with performance above a certain baseline; and

(2) online -- running best-arm-identification (BAI) algorithm (Audibert et al., 2010) and return the result.

In addition, there are two theoretical results:

Theorem 1 indicates that when the no. of models returned by CAS is large enough, we would find one that performs at least as good as the baseline model during online test (with high probability).

Theorem 2 analyzes the no. of iterations BAI (loosely speaking, how large it needs to be) requires to find an arm that performs better than the baseline (with high probability).

Finally, a synthetic experiment was provided comparing the proposed chaining of CAS-BAI vs BO-EI and UCB (initiated on random subset from the output of CAS). A more realistic ranking experiment is also conducted comparing CAS-BAI with BO.


**Broader Impact Concerns:**

No ethical concern

**Requested Changes:**

A few suggestions:

A more meaningful analysis need to capture how strongly must y and \tilde{y} correlate for this method to be effective. Such analysis can draw better insights that relate offline & online experiment. Otherwise, the current analysis is both fragmented & vacuous (see the above comment regarding theorem 1)

Theorem 2 needs more elaboration on what aspect is new given its similarity to UCB analysis.

The experiment section needs to be improved with more thorough comparison with other BO variants. From the description of the experiment, (1st sentence, 2nd paragraph, Section 4) it looks like BO finds a solution during offline phase but then it is not used during the online phase which is strange (according to table 2).

Please also include experiments with more realistic datasets and with larger scales.

Do consider provide the running time comparison too.

Btw, how is the content of 2.2 different from what is being criticized in page 2 -- "These strategies, however, are in pursuit of a continually/iteratively run online tracking/optimization process" -- it seems to do the same thing? Please do provide clarification.

**Strengths And Weaknesses:**

Strengths:

The paper is motivated from a realistic setting of model deployment, which is practical & interesting.
The writing is very clear and succinct.

Weaknesses:

To my understanding of this paper, the proposed algorithm runs CAS and then runs BAI to get the final results. Both existing algorithms are ran "as-is". With this, I think there is no algorithmic contribution. Please correct me if I misunderstand the paper.

Next, there are theoretical results but the results are disconnected. One result is specific to CAS setting while the other is specific to BAI setting. Both are established independently so again, this appears to be a loose couple of two separate works.

More critically, the theoretical results do not seem to convey any new insights at all. Going by the exact statement of theorem 1 -- when the no. of models selected (from CAS-constrained region) is larger than a constant, there will be one model which performs as least as good as a fixed baseline (with high probability) -- this is a vacuous result because assumption 2 already assumed that there exists one such model in the CAS-constrained region, so of course when the no. of randomly selected models become arbitrarily large, the probability of finding it can be arbitrarily high. Hence, theorem 1 did not prove anything that was not assumed away in assumption 2.

Then, theorem 2 is similar to UCB analysis so I am not sure what is new here.

Last, the experiment is too limited. First, the comparison with BO is severely lacking. There are many variants of BO but the comparison is restricted to EI -- in both synthetic & real-world experiment. The real-world experiment is particularly simplistic and uninsightful with 4 reported no. -- there can be many experiment scenarios with different x value for NDCG-at-x but only experiments with x = 1 were run. The BO should also have its various configurations but it seems the comparison is fixed to a single configuration.

Moreover, the scale of the experiment is too limited. The maximum no. of models that were advanced to BAI stage is no more than 5. But if that is the case, we can also deploy them all during the opportunity interval for online testing (Fig. 1) and select the best one? I am not sure why don't we go with that simple engineering solution?

---

> ### Author Response · Authors · 2022-08-19
> **Reply to mPxg**
>
> > To my understanding of this paper, the proposed algorithm runs CAS and then runs BAI to get the final results. Both existing algorithms are ran "as-is". With this, I think there is no algorithmic contribution.
>
> While it is true that we haven’t changed the expected coverage increase method (i.e., the algorithm proposed in Malkomes et al. (2021) to solve the Constraint Active Search problem), we do propose a novel algorithm inspired by the standard best-arm-identification algorithm. Our version is **customized** to identify “arms” (models) that are expected to yield better performance in the **online deployment phase**. Notice that we say “better” because we seek to find models that improve the application performance (online deployment phase) over the current baseline method in production, and we optimize the usage of our online stream pruning arms that are unlikely to improve the *known baseline desired performance*. We also consider the optimal number of arms K rather than giving it as an exogenous variable in our analysis.
>
> Most importantly, this work is not about a novel Constraint Active Search or Multi-arm-bandit algorithm, the novelty of our work comes from properly combining these ideas to address a common practical challenge. As far as we can tell, no other work has formalized this common ML-industry setting of developing and deploying online models, and no other MAB/BAI related paper has considered analyzing the settings where there is a known improvement target.
>
> > More critically, the theoretical results do not seem to convey any new insights at all. Going by the exact statement of theorem 1 (...)
>
> In Theorem 1, we identify the probability that at least one model (arm) performs as good as the fixed baseline ( K ~ Cmodel log(1/δ)). However, there is a tradeoff on how we choose $K$. The number of samples available in the online development phase is limited to T, and thus having more arms decreases the sample per an arm $T/K$ to compromise the accuracy of the estimation. Unlike the existing BAI literature where $K$ arms are exogenously given, our framework addresses the following question: **What is the best size of $K$ that we can find positive improvement over the baseline?** We address this in section 3.3 regarding the optimal choice of $K$.
>
> > Then, theorem 2 is similar to UCB analysis so I am not sure what is new here.
>
> Unlike standard UCB, this is the comparison with a fixed baseline and an algorithm is optimized for the probability of identifying an arm that is better than the baseline.
>
> > (...) First, the comparison with BO is severely lacking. There are many variants of BO but the comparison (..)
>
> * The point is not to demonstrate or investigate all the variants of BO. Our main motivation was to show that BO is designed to find a **single** best solution for the given objective function, which in practice is an offline metric. This becomes its downside when this solution has to be deployed in the online setting, where the online objective is different.
>
> * In fact, in Figure 2 we showed the best model found using the offline objective is often not the best model when deployed online (see the orange stars).  The BO method could be as arbitrarily good as this solution, but they would not be able to find the best “online” solution.
>
> * We picked “NDCG-1” as the online deployment objective as an easy way to mimic user behavior (binary feedback); given that
>    * We could not do multiple experiments using a real online recommendation system
>    * We think that understanding the experiment settings provides more insights on the pros and cons of the proposed methodologies as you correctly pointed out.
>
>
> In this circumstance, the offline and online metrics are correlated, but not perfectly. We agree that as x approaches 5 (as in NDCG-x), the offline and online metrics become more correlated and in that case using BO would become more advantageous. We will discuss this in more detail in our revision.

---

> > ### Author Response · Authors · 2022-08-19
> > **Reply to mPxg (cont)**
> >
> > > But if that is the case, we can also deploy them all during the opportunity interval for online testing (Fig. 1) and select the best one? I am not sure why we don't go with that simple engineering solution?
> >
> > Unfortunately we can’t deploy all of them at the same time. Each online response is a stream, in which a user requests a recommendation and the system needs to answer it with a response (recommendation).
> >
> > The whole point of performing the online deployment phase is to correctly select which arm (or model) we select to give recommendations. Different from normal multi-arm methodologies, our proposed method “prune arms”, which reduces the chances of serving out suboptimal arms.
> >
> > Choosing K as small as 5 is closely related to the practical setting where one wouldn’t maintain a large number of models to train and serve. Additionally, we have discussed the implications of choosing K theoretically, specifically in section 3.3. Notice, also, that the larger the K, the longer the online deployment phase needs to be to confidently identify which arm is likely to outperform the baseline.

---

### Review · Reviewer_uWZG · 2022-08-15

**Summary Of Contributions:**

This paper aims at tackling an interesting question that the performance can vary drastically between the online performance and offline validation setting, which is well-known in industrial ML settings. Concretely, this paper considers the setting where one have only a fixed window on which to conduct online development. This paper utilizes Constraint Active Search to identify a set of models, and studies their online performance using a variant of Best Arm Identification to select the best model for deployment. The authors claimed the main contribution of this paper is the theoretical analysis of the development phase.










**Broader Impact Concerns:**

This is no broader impact concerns from my review.

**Requested Changes:**

I have list all the questions I have in the above section. I hope the authors can answer my questions in detail and muffing the draft according to that.

**Strengths And Weaknesses:**

**Strengths:** This paper considers a very important question and the explanation of two-phase strategy is clear, and the theoretical analysis is correct from my point of view. I think this is a solid work but there are issues from my point of view.

**Weakness:**

1. In appendix B, how is DPP different from ECI? Could you explain (conceptually) which method might be better for $K$ model subselection in the offline deployment?

2. Will you analysis constrained by the assumption that $X(t)\sim Bern(y(x_{I(t)}))$? What will happen if reward has other distributions?

3. By assumption 3, it seems the quantity $\eta$ depends on $C_{Cov}$ and the definition of $\eta(r)$ should be $\eta(r,C_{Cov})$?

4. As mentioned in remark 4, the assumption of $x_i$ being i.i.d seems to largely reduce the difficulty of the analysis, can some martingale technique can be leveraged for showing the non-iid case (since the authors claim the biggest contribution is the theorem)?

5. How would the $\eta$ quantity in Assumption 3 affect Theorem 1 here? If $\eta$ will not affect Theorem 1, can you provide some explanation why this is the case?

6. The numbers in Table 3 seems to be close for BO and this paper. How to verify the result in Table 3 is statistically significant?

7. You have two-stage strategy, can you explain which stage of your design plays a more critical role for better performance (comparing to BO)?

---

> ### Author Response · Authors · 2022-08-18
> **Reply to reviewer uWZG**
>
> > 1. In appendix B, how is DPP different from ECI? Could you explain (conceptually) which method might be better for  model subselection in the offline deployment?
>
> In theory the DPP subselection strategy would be a better candidate for approximately selecting dispersed models throughout the satisfactory region S. Recall that in Theorem 1 we have utilized the fact that our samples are i.i.d.
>
> Greedily maximizing ECI as we have proposed, also provides a set of diverse models on S, but in different fashion, arguably a more conservative one. First, this subselection scheme will be less likely to favor points close to the boundaries of the satisfactory region S due to the coverage criteria. And secondly, we always add the highest observed value first. Both choices favor selecting points in the "interior" of the satisfactory region.
>
> If practitioners expect a considerable discrepancy between offline and online settings, then the DPP strategy will likely increase the odds of finding a model with better performance than the baseline. Otherwise a conservative choice such as sub selecting models using the greedily ECI approach will give a good balance between diversity and high performance.
>
> We will further clarify this on appendix B.
>
> > 2. Will you analysis constrained by the assumption that ? What will happen if reward has other distributions?
>
>
> We considered the Bernoulli case because it represents the simplest engagement of people in web application. It is easy to extend our results with other distributions such as sub-Gaussian.
>
>
> > 3. By assumption 3, it seems the quantity  depends on  and the definition of  should be ?
>
> You are right that $\xi$ depends on $C_{cov}$. We clarify this and omit $C_{cov}$ when it is clear. Thank you for your clarification.
>
>
> > 4. As mentioned in remark 4, the assumption of  being i.i.d seems to largely reduce the difficulty of the analysis, can some martingale technique can be leveraged for showing the non-iid case (since the authors claim the biggest contribution is the theorem)?
>
> It is true that the sampling used in the simulations is not i.i.d. but k-DPP (Kulesza and Taskar 2011). This is because k-DPP covers the volume more uniformly than i.i.d. sampling (Figure 1 in Kulesza and Taskar 2011), which results in a better practical performance.
>
> We consider an analysis of (k-)DPP to be very interesting but is out of the scope of this paper. First, the uniformness of DPP has not been analyzed well in the machine learning community (at least not covered in the popular book by (Kulesza and Task 2012)). The uniformness property we need in our theoretical analysis (for deriving similar results to our Theorem 1) is called the "gap probability" of DPP. While the gap probability of DPP is given as an expansion series (Girotti 2011), evaluating such a series is mostly done in the more basic mathematics community, which we believe to be too much for this paper.
>
> - (Kulesza and Task 2012) "Determinantal Point Processes for Machine Learning" Alex Kulesza and Ben Task
> - (Girotti 2011) "Random Matrix Theory in a nutshell Part I: Determinantal Point Processes." Manuela Girotti.
>
> > 5. How would the  quantity in Assumption 3 affect Theorem 1 here? If  will not affect Theorem 1, can you provide some explanation why this is the case?
>
> This assumption avoids some corner cases. Figure 3 right shows such a case, in this case, the probability that an i.i.d. sample is drawn from the sharp spine is very small, and if the improvement region $\mathcal{S}_{imp}$ is limited to the spine, then the i.i.d. samples do not work. This assumption is rather theoretical and we believe this does not matter in practice.

---

> > ### Author Response · Authors · 2022-08-18
> > **Reply to reviewer uWZG (cont)**
> >
> > > 6. The numbers in Table 3 seems to be close for BO and this paper. How to verify the result in Table 3 is statistically significant?;
> > 7. You have two-stage strategy, can you explain which stage of your design plays a more critical role for better performance (comparing to BO)?
> >
> > Our proposed two-stage pipeline represents a prevalent ML-industry setting and it is important to highlight what we mean by performance. Ultimately, we wish to improve our **online deployment phase** (i.e. your application performance online).
> >
> > With this particular goal in mind we must stress that a strict comparison with BO is impossible. Specifically, a standard BO pipeline consists of selecting the best offline model and using it online. If the offline and online settings are different, BO suffers from a significant loss even if we pay an infinite cost for the offline experiments.
> >
> > Our two-stage pipeline is different, and tailored to online ML applications. We select multiple (let's say K) offline models, we evaluate them in an **online development phase**, and we select the model most likely to *improve* our **baseline online deployment performance**; where the *baseline online deployment performance* is measured by our current model in production.
> > Perhaps, it is easier to contrast our two-stage approach with the standard A/B test, which is commonly used to estimate the online deployment performance of two models:  the best offline model (for example the BO selection) against the current model in production (baseline model). Our approach is a *generalization* of the A/B test since it allows an arbitrary comparison among K models; but it is also *customized* to the online development/deployment cycles which focus on improving the performance over a **known** baseline model.
> >
> > Recall that in our set of synthetic experiments (Section 4.1), we gave evidence to support our hypothesis that our proposed methodology is more robust against discrepancies in the offline/online settings; whereas BO fails to account for this discrepancy. That is, *we expect better performance when compared to BO if the offline/online settings are different*. (See Table 1 as the magnitude of the shift increases). Naturally, and as expected, BO will excel if the two settings are extremely correlated (first column Table 1); in that case the standard Bayesian optimization is "optimal" and more simple (single model, easier pipeline).
> >
> > With this context in mind, we can interpret the results from Table 3 again. Here, our goal is to provide evidence on two real datasets which we have no control over the data generation process. Figure 2 shows the correlation between the offline and online metrics, and based on that plot we can hypothesize that our strategy will be more effective on the dataset MQ2008 and comparable to BO on the MSLR-WEB10K dataset. We have increased the number of repetitions for this experiment and we are going to update Table 3 and the text accordingly. Our results for the dataset MQ2008 are statistically significant at a significance level of 0.001 according to a two-sided t-test. Our current results for MSLR-WEB10K are not statistically different so we cannot reject the null hypothesis that they have the same performance.

---

### Review · Reviewer_Mp3i · 2022-08-15

**Summary Of Contributions:**

The authors study the problem of training a set of models using an offline dataset for the purpose of selecting a model for online deployment. The offline data may not come from the same distribution as the online test will be. They introduce an algorithm with two phases, one that produces using the offline data a diverse set of models via constraint active search, and a second one that tests the models via a bandit algorithm on the test data distribution. The best model (or models) surviving the bandit best arm identification algorithm are the candidates for deployment at test time. The authors present a few simple theoretical snippets that justify their approach and conduct simple experiments in synthetic and recommender system datasets comparing the performance of their algorithm with other benchmarks.

**Broader Impact Concerns:**

This work does not pose any adverse societal consequences.

**Requested Changes:**

The algorithms are extremely under specified. It is very hard to understand what are they. The paper lacks an explanation on constraint active search. This discussion should be in the main part of the paper. There should be an algorithm box for the main methods and not only a vague textual description. The experimental evaluation is very unclear. There is no description of what the baseline does. This work cannot be considered for acceptance without a major rewrite that enhances clarity, lets the reader understand what is the exact algorithm the authors are proposing and what are the baselines they are comparing against. Addressing concern 1) in the 'strengths and weaknesses' section would be ideal.

**Strengths And Weaknesses:**

The paper studies an interesting problem and makes a good argument in the introduction as to why we should care. The observation that offline development and online deployment may happen in different data distributions. The ideas proposed by the authors to mitigate this issue are intriguing. It makes a lot of sense to produce diverse models and select the best among them for test. The overall strategy for producing these diverse models is interesting, and the use of DPPs for diversity selection is a great idea.

That being said, the paper has multiple flaws. The main issue being clarity. There is no algorithm box for the whole algorithm, which makes it very hard to understand the precise mechanism. There is no explanation of what constraint active search is. It is therefore unclear how that is used to produce the different models in the offline phase. The theorems are underwhelming as they are mere corollaries of existing stuff, or very simple rehashing of existing results. For example, Theorem 1 is just a result about repeated trials and how the probability is boosted exponentially of seeing a rare outcome. Theorem 2 is an inverted Hoeffding bound. The exponential convergence of remark 7 is a simple inversion of the Hoeffding bound formula.

I will list other problems

1) If the learner has access to  a small online dataset for the online development phase, can't she simply use all this data to decide what model is best? The best arm identification bandit procedure is lossy. If it doesn't and this small dataset of test data is accessed online, why should the learner need to stop exactly after $T$ rounds?

2) The BO baseline is not explained? Is it two step? It is completely unclear.

---

> ### Author Response · Authors · 2022-08-24
> **Reply to Reviewer Mp3i**
>
> Thanks for your contribution; we will add the description of the algorithms with proper environments to improve clarity in our revision. With respect to your questions:
>
> * (First question 1) It's important to understand that we are not considering a standard supervised learning setting, so the notion of *a small dataset* from your question does not apply here. The data is an online stream that needs to receive an *online recommendation*. Our data cannot be used twice, and we do not have counterfactuals (we don’t know the user’s response if we had served them a different recommendation).
>
> * (Second question in 1) And we have to properly manage the exposure of our system to online recommendations (that's why we have a limit on the number of rounds); in other words, we have to be careful about serving recommendations of arms with unknown performance. So the T rounds represent a budget or a total cost that we are willing to pay for experimentation.
>
> * (Question 2) The BO baseline mimics what most practitioners do today: find the best offline model and test it online. If the *estimated online performance* is better than their current production model, they deploy the new model (arm).
>
> Concerning the requested changes, we agree that we could further improve clarity for *not-so-code-inclined* readers by adding a more detailed description of ECI. We respectfully disagree with the review regarding *underspecification of our algorithms* and the reproducibility of our experiments. To the best of our knowledge, Section 4, Appendix E, and F have *all the information needed to replicate our results*. Specifically, we included a link to the implementation (a tutorial) of ECI that we used in our experiments, as explained in Appendix F.2 (Page 15) footnote three; the same library was used to implement Expected Improvement as the BO baseline. The only “omission” on our part is the implementation of UCB, which we considered obvious and/or not central for most readers.

---

### Comment · Action_Editors · 2022-08-15
**Additional Comments**

Dear Authors,

I am the Action Editor for this submission -- I opted to also read over this paper myself, and I give some comments below.  Some of them were already mentioned by the reviewers, but I will keep those listed as well to highlight that they arose independently from multiple readers.

You can treat this as a "4th review" and edit/respond accordingly.

Regards,

Area Chair.

===

My main comments are on some of the mathematical steps:

i)	At the start of Section 2, please fully describe the sequential decision-making process.  For example, in BO the process would be to sequentially select $x_t$ and observe $y_t = f(x_t) + z_t$ with Gaussian $z_t$.  You can similarly be more explicit to make your setup as clear as possible.

ii)	At various stages it’s said that y(.) is modeled as a GP, but this may cause problems at various points.  First, a GP will have a probability of falling outside [0,1], so Bernoulli(y(.)) in (1) will not make sense.  Second, Assumption 2 becomes questionable, because you would be placing a deterministic assumption on a random function (so it will fail with some probability).  Overall, I think things will be neater if you just drop the GP assumption.  Then instead of Assumption 1 you can just have a deterministic assumption on the Lipschitz constant.

iii)	What Theorem 1 doesn’t capture is that taking i.i.d. samples from S requires first finding S itself, which may be difficult.  While it’s not necessary to characterize this (previous works on level-set estimation may already do so), this point should be highlighted.

iv)	(Minor) In Remark 7, I was confused as to why $\pi^2$ is suddenly introduced.  On the same page, should the second-last display equation be an inequality?

v)	I found the bottom of p6 potentially misleading, because my understanding is that Delta is both a random quantity and dependent on K – though perhaps under reasonable conditions it can be treated as a constant for these purposes.

vi)	The application of Assumption 3 in Appendix C could be clearer.  Here you established something about a ball around $x^*$, whereas Assumption 3 is about a ball around various $x_i$’s (which may all differ from $x^*$).  I suggest taking this step more slowly and giving a bit more hint.

vii)	Also in Appendix C, once you prove that S_imp contains a ball of radius D/L, why not just use that to lower bound Vol(S_imp) directly, without requiring Assumption 2?  Is it because you are seeking a better final bound?  On a similar note, perhaps the theorem should be stated with an explicit bound on K to make it more insightful.


Some less mathematical comments are as follows:

a)	If I understand right, the first phase gets regular BO-like observations but the second phase gets Bernoulli observations.  This seems a bit unusual, and should be commented on if the paper keeps this setup.

b)	On p2, it’s suggested that previous work is limited because it requires sequential online optimization in the second stage, whereas you require your own “process to be fully fixed”.  I did not understand this, as your second stage looks like a sequential online optimization strategy (UCB) as well.

c)	In Section 2.1, it’s stated that you use CAS, but the reader may not be familiar with that.  The reader can also get confused by consulting this reference, because it’s a multi-task paper and yours is not.  Can you include a LaTeX algorithm environment to give a self-contained description?  (Possibly in an appendix). It should be particularly  clear what CAS takes as input/output, its sequential/adaptive nature, etc.

d)	On p4 (“…more than one…”): I believe some works have considered the problem of finding just one good arm, e.g., see “The True Sample Complexity of Identifying Good Arms” and possibly other references therein.

e)	Assumption 3 reads more like a “definition” than an “assumption”, and would benefit from re-wording.

f)	If the paper only considers noise-free observations, then I suggest at least mentioning noisy settings for future work in the Conclusion section.

---

> ### Author Response · Authors · 2022-08-26
> **Thank you for many suggestions. We answer each of the questions below.**
>
> i) Thank you for your suggestion. We will clarify each of the two sequential processes (offline development phase and online development phase, Figure 1). In particular, the offline development phase sequentially selects the model with *all training data* ($\tilde{y}(x)$ is the performance of the model x against all training data), whereas the online development phase sequentially selects the model and evaluate it with *single user* (X(t) is feedback from a single user).
>
> ii) The difference of the feedback (GP in the offline development phase and Bernoulli in the online development phase) stems from the difference of the two phases (our answer to point i) ). We can relax the assumption in the offline development phase. We don’t need to assume that y(.) is drawn or modeled as a GP, a Lipschitz assumption suffices. Thank you for pointing it out. In our experiments, we only use GPs during the offline phase.
>
> iii) We will highlight this after presenting Theorem 1. Recall that our two-phase algorithm only requires the satisfactory level \tau, i.e. we already incorporate “finding S” via adaptive sampling (approximately) solving the offline Constraint Active Search problem.
>
> iv) pi^2 is introduced to bound \sum_{n=1}^\infty 1/n^2 <= pi^2/6. We will fix the inequality in accordance with your correction.
>
> v) You are correct. Delta should be marginalized with respect to Bayesian prior / given high-probability lower bound. If Delta > 0 with some constant probability, then our high-level discussion that K = O(\sqrt{T}) holds. We will add discussion on this for our revision.
>
> vi) We will add additional detail on Appendix C in accordance with your suggestion.
>
> vii) You are correct that we derive Vol(S_imp) from the packing number, and subsequent results rely on this quantity. We start from the standard Gaussian process assumption to derive Vol(S_imp), but will add discussion on this.
>
> a) That is a correct understanding. We hope that our new setup summary makes this distinction clear. We will make a post with all the updates to the manuscript.
>
> b) We will improve the main text to clarify this, but what we mean to say is that Letham & Bakshy considered a more sophisticated scenario, where offline and online observations can be collected in multiple successions.  In general, we do not enjoy such luxury of successive alternating between offline/online experimentation using simulators.
> We considered a different setting that in our opinion captures a more standard industry scenario where developers build offline models and somehow infrequently test these models online. We assume that the cost of online experimentation is expensive and can only be performed during a short period of time, where system administrators have to redirect the online traffic to perform experimentation. Notice that even if the operation aspect of managing online traffic is automated, there is an important business decision of not serving users with your current production model, which invariably comes with the risk of sacrificing performance.
>
> c) That’s a good suggestion; we will add this information to the Appendix, thanks for bringing it up
>
> d) While the algorithm in “The True Sample Complexity of Identifying Good Arms” is more general than ours, it involves phases (l=1,2,...) where the set of active arms in early phases is limited in the early phases for the cost of generality. Our algorithm is aimed for our aim and does not involve such a phase, and as a result much simpler. We consider using the algorithm that is designed for the problem is important for a practitioner because it is easy to implement and does a more predictable behavior, even though there is a more general algorithm that behaves similarly when delta -> 0.
>
> e) We will consider re-wording it. Initially we left it as an assumption to get rid of the degenerative case where there is only one point isolated (in this case, C_{cov} = 0)
>
> f) We make no structural assumption about how $y$ and $\tilde{y}(x)$ are correlated, so the offline metric might be any corrupted version of the online one. The user’s online feedback must be samples from the expected reward distribution.

---

> > ### Comment · Action_Editors · 2022-09-25
> > **Requests**
> >
> > Dear authors,
> >
> > I'm still a bit confused about Assumption 3.  Could I request the following?
> > 1) Reply here (or update your PDF) with how you will re-word the assumption to incorporate language to the effect of "...we assume..." and hence clearly distinguish what is "assumed" vs. what is just a "definition"
> > 2) More importantly, please clearly/rigorously show how Assumptions 1 and 3 combine to give Equation (3) in Appendix C.  I was initially confused about the difference between $x_i$ and $x^*$ (see item (vi) in my previous post), and in the revision there is even less detail around (3) and Assumption 3 is not mentioned.  (Perhaps some text was accidentally removed.)
> >
> > Sorry that this request is coming late, but I will proceed with the decision ASAP afterwards unless follow-up questions are needed.
> >
> > Thanks.

---

> > > ### Author Response · Authors · 2022-09-27
> > > **revised the paper**
> > >
> > > Dear AE,
> > >
> > > We appreciate your careful reading. Below are our answers.
> > > * Our assumption is that the model space satisfies $C_{cov} \ne 0$ (and thus $C_{cov} > 0$).
> > > * Thank you for pointing it out. The fact that the covering does not include $x^*$ causes an issue, which is fixed by taking a cover of r=D/(2L) instead of D/L. We corrected the proof accordingly and revised the paper.

---

### Author Response · Authors · 2022-08-18
**Thanks for all your reviews!**

We would like to thank you all for your great feedback! We all have been busy with ICML and NeurIPS, so we appreciate all your effort and time. We are going to be making individual comments to each reviewer this week and we plan to update the manuscript with all the updates next week. Happy to provide further clarification as needed.

---

### Author Response · Authors · 2022-08-30
**New version uploaded**

All major changes are highlighted in blue in the new version of the manuscript. Here is a short summary of the main changes after the reviewers contributions.

- Clarification about differences with Letham & Bakshy (2019)
- Summary our our problem formulation, beginning of section 2
- Better discussion of the differences of our online algorithm
- Replace previous assumption with just Lipschitz continuity
- Experiments ran for longer, 100 repetitions
- New experiment with a corrupted version of the dataset MSLR-WEB10K
- New results Table 3, and appropriate discussion
- New Appendix A with detail description of Constraint Active Search with Expected Coverage Increase

---

### Comment · Action_Editors · 2022-09-15
**Reviewer Recommendations**

Dear **reviewers uWZG and Mp3i**:

Please submit your final recommendations ASAP, as they are now overdue.  Thanks.

---

### Decision · Action_Editors · 2022-09-27

**Recommendation:** Accept with minor revision

**Comment:**

While all recommendations are in the "leaning" rather than "strong" category, the consensus is that generally that the concerns were addressed adequately, and the paper meets the main criteria of being correct and of interest to a subset of the TMLR audience.

Some remaining concerns (but not cause for rejection) included limitations in the theoretical results (including the iid assumption) and possible further presentation improvements.

For the final version, please proof-read the newly-added (blue) parts, and perhaps the whole paper, for minor mistakes or better wordings.  For example:
- After Eq. (3), the authors introduce V as being a subset of S-imp, but then need some mathematical working to establish that this is true.  The "subset of S-imp" part after "Let V..." should be removed.
- Assumption 3 should at least end with "We assume that C_cov > 0.", otherwise none of the language actually says anything about assuming anything.  (Such language is only used *after* the formal assumption, not within it.)  In fact I think the whole assumption should be re-worded for better clarity, as it still reads like a definition (of \xi).  Finally, it's probably worth highlighting that C_cov doesn't depend on r, maybe even be rewording the assumption like "We assume there exists a positive constant C_cov > 0 such that for all r...".
- In Appendix D you need to mention Assumption 2, not just 1 and 3
- In Appendix D please give a hint why V1 is a subset of V (triangle inequality?)
- Check for typos (e.g. "for most shape[s]")

On reviewer suggested that the notation could be modified since x often represents fixed data inputs (which one has no control over), but I believe this change should not be necessary.

**Audience:**

The findings should be of interest to readers working on Bayesian optimization, adaptive experimentation, and related topics.

**Claims Support:**

The claims are generally sufficiently supported.  The theorems are backed up by rigorous proofs, though the reviewers partially questioned the significance of the theorems themselves.  The experiments could be broader in scope, but are generally sufficient.

**Main Claims:**

The main claims/findings are roughly as follows:
- Sequential experimentation problems can benefit from combining an offline phase (to find a diverse set of models to consider) with an online phase (to narrow those models down to a best one)
- Under certain assumptions, the proposed algorithm has useful theoretical guarantees for each phase
- Experimentally, the proposed method can outperform classical approaches such as pure Bayesian optimization

---

> ### Comment · Action_Editors · 2022-10-20
> **Assumption 3**
>
> Dear authors,
>
> I need to ask you to edit Assumption 3 again.  I still believe there are at least two problems:
> 1) The first sentence makes C_cov sound like a fixed constant (in particular not depending on r), but the display equation writes "C_cov = (expression depending on r)".  These don't seem consistent.
> 2) I still think the current wording is too unclear in distinguishing the definition of C_cov vs. the definition of \xi (the two are somehow inter-twined and confusing or maybe circular).  As written, it looks like I could set C_cov = 1 even in your Appendix C counter-example, and Assumption 3 would still "hold", just with \xi(.) = 1, which is trivial (and gives log(0) in your analysis).
>
> This assumption is central to your analysis, and I can only do a final approval once I'm convinced the assumption is stated correctly.  Please reply here once you have edited, or alternatively, if you find yourself disagreeing with any of the above.

---

> > ### Comment · Action_Editors · 2022-10-20
> > **Follow-Up**
> >
> > Actually, I'm not sure if the exact numbers with C_cov=1 and xi(.)=1 in my previous post are correct -- but I hope the message of my general confusion is clear.  There may even be further issues with the order of quantifiers beyond what I mentioned, because writing "For any point x let \xi..." suggests that \xi is allowed to depend on x.  Overall, I believe that a complete re-wording / re-writing of the assumption is probably in order, and I urge taking as much care as possible in doing so.

---

> > > ### Author Response · Authors · 2022-10-20
> > > **Revision on Assumption 3**
> > >
> > > Dear Action Editor,
> > >
> > > Thank you for your careful reading that helps us to improve the paper. We rewrote Assumption 3 as well as experimental section and appendix related to offline development phase (CAS). Regarding Assumption 3, we made $C_{\mathrm{cov}}$ as a function of $r$.
> > > * Indeed, it depends on the choice of $\xi(r)$. It does not depend on a particular point $x \in \mathcal{S}$.
> > > * Making $C_{\mathrm{cov}}(r)$ as a function of $r$ does not rule out the possibility of the limit $\inf_{r>0} C_{\mathrm{cov}}(r) = 0$ (for a sequence of $\xi(r)$). However, we think that is not an issue because (1) $\inf_{r>0} C_{\mathrm{cov}}(r) > 0$ for standard shapes such as hypersphere and hyperrectangle, and (2) we only use one particular $r = D/(2L)$ that depends on $D$ (amount of improvement) and $L$ (Lipschitz continuity of the objective) - if $D/L$ is very small, then it implies the improvement is very small or the objective function is very sensitive to in the feature space of $x$, in that case, we should rather rethink modeling anyway.
> > >
> > > We are happy to rewrite further should there be any space for improvement.

---

> > > > ### Comment · Action_Editors · 2022-10-21
> > > > **Thanks**
> > > >
> > > > Thanks, this seems to be sufficient.  However, please re-upload with the typo in (6) fixed -- the denominator should be -log(.) or log(1/(.)) instead of log(.).  While you're at it, you might as well include the following very minor changes: (i) Add "the" after "See Appendix D for", and (ii) Throughout Appendix D, replace C_cov by C_cov(D/(2L)).

---

> > > > > ### Author Response · Authors · 2022-10-21
> > > > > **Thank you for the corrections**
> > > > >
> > > > > Dear Action Editor,
> > > > >
> > > > > We corrected the mistakes accordingly. Thank you again for your suggestions.